# Exploratory randomised trial of face-to-face and mobile phone counselling against usual care for tobacco cessation in Indian primary care: a randomised controlled trial protocol for project CERTAIN

Rajmohan Panda,[1] Rumana Omar,[2] Rachael Hunter ,[3] Rajath R Prabhu,[1] Arti Mishra,[4] Irwin Nazareth [3]

¹Research, Public Health Foundation of India, Gurgoan, Haryana, India
²Department of Statistical Science, University College London, London, UK
³Department of Primary Care and Population Health, University College London, London, UK
⁴Research, Independent Consultant, New Delhi, India

**Correspondence to**
Dr Rajmohan Panda;
raj.panda@phfi.org

## ABSTRACT

**Introduction** Despite widespread use of smokeless tobacco products by people within the Indian subcontinent, there is little awareness among Indians of its health hazards when compared with smoked tobacco. We hypothesise that mobile phone counselling will be feasible and effective for smokeless tobacco cessation intervention in India. This paper presents the protocol of the development and conduct of an exploratory trial before progression to a full randomised controlled trial.

**Methods and analysis** An exploratory randomised controlled trial will be conducted in urban primary health centres in the state of Odisha, India. A total of 250 smokeless tobacco users will be recruited to the study (125 in each arm). Participants in the intervention arm will receive routine care together with a face-to-face counselling intervention followed by advice and reminder mobile messages. The control arm will receive routine care, delivered by a primary care physician based on 'Ask' and 'Advice'. All participants will be followed up for 3 months from the first counselling session. The primary outcome of this trial is to assess the feasibility to carry out a full randomised controlled trial.

**Ethics and dissemination** Ethical approvals were obtained from the Institutional Ethics Committee of Public Health Foundation of India, Health Ministry's Screening Committee, Odisha State Ethics Board and also from University College London Research Ethics Committee, UK. The study findings will be published in a peer-reviewed scientific journal.

**Trial registration number** CTRI/2019/05/019484.

## Strengths and limitations of this study

► Evidence on the effectiveness of smokeless tobacco (SLT) cessation intervention in primary care is scarce. The results of this study will inform on the feasibility of conducting an effectiveness trial in the future.
► Randomisation with allocation concealment, blinded assessment of outcome measures and registration of the protocol in the trial registry will minimise potential bias.
► Providing training to health professionals will lead to an improved understanding of the delivery of low-cost tobacco cessation interventions in primary care.
► The primary care physicians may get unmasked to the allocation of the participants during the course of the trial as the participants visiting the urban primary health centres may reveal the information to the physicians. There is a chance of biased results if physicians may deliver repeated face-to-face counselling services.
► The inclusion criteria include SLT users having a mobile phone, this may exclude a section of the SLT users who do not have a personal mobile phone like very poor or the elderly who do not use a mobile phone.

mixture of cultures, religions and practices. The majority of the tobacco users in India use a variety of tobacco products—combustible, non-combustible or both. As per the Global Adult Tobacco Survey 2 (GATS 2), 28.6% of the adult population in India consumes tobacco (10.7% of smoking and 21.4% of smokeless), making it the second-largest consumer in the world.[3]

Widespread use of smokeless tobacco (SLT) products occurs in countries such as in

## INTRODUCTION

Tobacco use is responsible for almost eight million deaths each year or a death every 6 seconds.[1 2] About 80% of 1.3 billion tobacco consumers across the globe live in low and middle-income countries (LMICs).[1] In LMICs like India, the tobacco problem is complex as the country has a diverse population with a

India and Bangladesh accounting for 232 of 248 million SLT users worldwide (i.e., from 21 countries).[4] SLT products such as *tambaaku*, *gurkha*, *zarda*, *khaini*, *mawa* and *pan masala* are widely used in many states in India as SLT consumption is culturally accepted and is considered common practice.[5] In India, SLT use is double that of combustible tobacco, making it the single most common form of tobacco used by men and women (29.6% and 12.8%, respectively).[5] According to GATS 2 survey, the state of Odisha was one where the largest proportion of adults consumed SLT products (43%) compared with the national average (21.4%).[6] Despite strong evidence of SLT[7] on cancer of the oral cavity, pharynx, oesophagus and also responsible for a large proportion of tobacco-related cancers in India, there is a misconception that SLT is relatively harmless .[8]

Primary care health professionals, the first contact for patients accessing help, play a key role in the health education of the hazards of tobacco. They are best placed to counsel against tobacco use and promote cessation. Studies on brief interventions by healthcare providers are effective in motivating tobacco users to quit tobacco.[9 10] A systematic review on SLT cessation intervention from both high-income countries (HICs) and LMICs showed that behavioural interventions are effective in achieving cessation.[11] However, the overburdened primary care physicians in India have limited time for counselling,[2 12 13] and it is also difficult to get users to attend follow-up visits as journeys to primary care clinics would incur a loss of income.[12]

In India, mobile phones are widely used in both rural and urban settings. India with 1173.7 million (98.2%) mobile phone users is the second-largest user base in the world.[14] Mobile phone subscribers in India increased from 1187 million in June 2019 to 1195.2 million by September 2019.[14] A meta-analysis of 104 studies including randomised trials and quasi randomised trials on the effectiveness of telephone counselling for smoking tobacco cessation showed significant but modest effect sizes in HICs.[15] Effectiveness increases with treatment intensity.[16] A review of 26 trials on smoking cessation demonstrated that automated text messaging interventions were more effective than minimal smoking cessation support (RR 1.54, 95% CI 1.19 to 2.00; $I^2$=71%; 13 studies) and text messaging added to other smoking cessation interventions was more effective than other individual smoking cessation interventions (RR 1.59, 95% CI 1.09 to 2.33; $I^2$=0%, four studies).[17] Another meta-analysis of 13 trials reported smoking quit rates with text messaging intervention were 35% higher than quit rates for controls (OR=1.35, 95% CI 1.23 to 1.49).[18] Other reviews of studies from HICs suggest that such interventions can increase the chance of quitting smoking tobacco from 39% to 80%.[19 20] Studies on SLT cessation that included randomised controlled trials conducted in HICs as well as some LMICs showed that behavioural cessation interventions led to quit rates between 9% and 51.5% at 6 months.[11 21] There are, however, little data on the efficacy and effectiveness of such intervention combined with mobile phone-based counselling in LMIC like India. Improved participation in tobacco cessation programmes through the use of mobile phone technology can have a substantial impact on a population at risk even with a small effect size.[22]

There is an urgent need to test the effectiveness of the use of mobile phone messages for SLT cessation in the Indian primary care setting.[23] We hypothesise that a face-to-face intervention offered by a tobacco counsellor together with low-cost mobile-based counselling has the potential to deliver a high-quality cessation intervention to SLT users. Prior to that, we need to test the feasibility and acceptability of recruitment, delivery of the intervention and the follow-up of participants within a randomised controlled trial.

## AIM AND OBJECTIVES

Our overarching future aim is to evaluate within a randomised controlled trial, the clinical effectiveness (as measured by tobacco cessation) and cost-effectiveness of a complex intervention of face-to-face counselling coupled with mobile phone messaging delivered to SLT users visiting Indian primary care clinics in addition to routine care provided by the primary care professional against routine care alone. Prior to undertaking a full trial, we aim to develop and finalise the intervention and then test its acceptability and feasibility. Specific objectives of this study are to assess within an exploratory randomised controlled trial, the proportion of:

1. ll primary care SLT user attendees approached to take part in the trial that consent to randomisation.
2. Those who are randomised to the intervention comply with the intervention.
3. Those who are randomised on whom the follow-up data can be collected.
4. Missing data on all research measures administered at baseline and 3 months, which will be used to assess the primary and secondary outcomes for conducting effectiveness randomised controlled trial in the future.

## METHODOLOGY
### Patient and public involvement
The patients and public were involved in the formulation of the study hypothesis from previous research studies.[24 25] The research questions, design and outcome measures were derived from previous interaction with different stakeholders including patient and public. Tobacco users and other stakeholders (including primary care physicians, counsellors) will be involved in the development of two components of the intervention, that is, face-to-face counselling and mobile phone messages. These will be done by in-depth interviews and focus group discussion in the formative phase of the study (online supplemental appendix A). The summary of the results of the study will be made available to patients and public at the end of the project period.

**Table 1** SPIRIT figure illustrating the phases of CERTAIN trial and data collection time points

| | Study period | | | | | |
| --- | --- | --- | --- | --- | --- | --- |
| | Pre-enrolment | Enrolment | Allocation | Post allocation | | Close-out |
| Time point | 0 | 0 | 0 | 0 | 3 months | After 3 months |
| **Enrolment** | | | | | | |
| Eligibility screening | | X | | | | |
| Informed consent | | X | | | | |
| Randomisation to treatment allocation | | | X | | | |
| **Interventions** | | | | | | |
| Routine care | X | | | | | |
| Ten-minute face-to-face counselling | | | | X | | |
| Mobile message-based counselling | | | | | X | |
| **Assessments** | | | | | | |
| Demographic | | X | | | | |
| Baseline assessment | | X | | | | |
| Mid-line assessment—qualitative assessment at 1.5 months of recruitment | | | | | X | |
| End-point assessment | | | | | | X |
| Saliva cotinine assessment | | | | | | X |
| Qualitative assessment with drop outs | | | | | | X |
| Qualitative assessment with participants who successfully completed the follow-up | | | | | | X |

CERTAIN, Counselling intErvention foR smokeless Tobacco cessAtion in INdian primary care; SPIRIT, Standard Protocol Items: Recommendations for Interventional Trials.

## Study design

The study will be conducted in two different phases. The first phase is a formative phase (online supplemental appendix A) and the second phase is an exploratory, parallel-group, randomised controlled trial, with pretest and post-test assessments. The duration of the study will be 18 months. The first phase will last for 8 months, whereas the second phase will occur over 10 months (see table 1). The study flowchart is provided in figure 1.

## Study setting

The study will be conducted in urban primary health centres (UPHCs). All UPHCs in Berhampur city of Ganjam district of Odisha will be selected for this study.

## Study population

The study participants will include adult SLT users who visit UPHCs in Berhampur city, Odisha. Participants will be current SLT users (i.e, users in the last 3 months), aged 18 years and above, have a mobile phone with a valid contact number and willing to consent to participate in the study.

Participants below the age of 18 years, who do not have the mental capacity to consent, with illness limiting their adherence and follow-up within the study, will be excluded.

## Interventions

The intervention arm includes routine care (component 1) along with a single 10-min face-to-face intervention based on 5A's approach (component 2), followed by mobile phone counselling (component 3). The control arm will be the provision of routine care alone (ie, component 1).

Counselling intErvention for smokeless Tobacco cessAtion in INdian primary care (CERTAIN) intervention: the intervention will be a combination of the following components

1. Component 1 or delivery of 'routine care' as delivered by a primary care physician over 1 min to 2 min. This component will be delivered to both the intervention and control arms of the study. This is based on 'Ask' and 'Advice'. Under 'Ask', the physician will ask about consumption of SLT and under 'Advice', the physician will advise participants about the benefits of complete abstinence in a clear personalised manner.

2. Component 2 will include a single 10-min face-to-face intervention delivered by a practice-based counsellor. This component will include brief standardised advice provided to participants based on 5A's approach[26] to tobacco cessation. Under this approach, the counsellor will 'Ask' the participants whether they use SLT; 'Advice' them on the importance of quitting (following the same script of component 1). Additionally, they

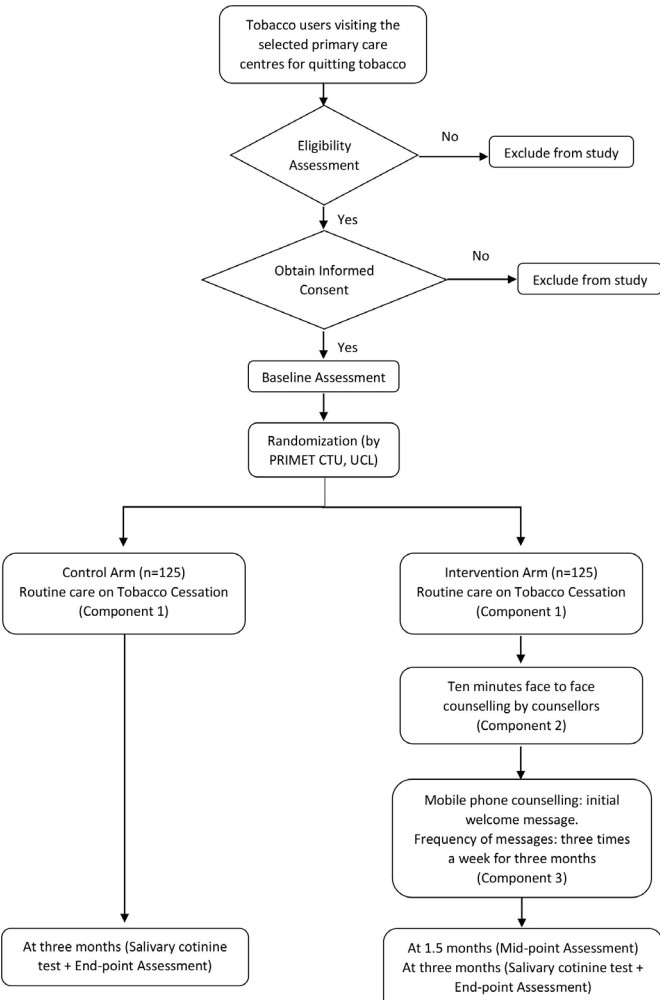

**Figure 1** Trial schema showing the pathway for patients' recruitment process in Counselling intErvention foR smokeless Tobacco cessAtion in INdian primary care (CERTAIN) study.

will 'Assess' their willingness to 'quit now'; offer them 'Assistance' in the form of nicotine replacement therapy and/or provide them with a referral for behavioural support and 'Arrange' a follow-up to check on progress. This will conclude with their enrolment to receive component 3—mobile phone intervention.

3. Component 3 will also follow 5A's approach to tobacco cessation. The intervention will include an initial message after 48 hours of recruitment, to remind them to quit tobacco followed by messages occurring three times a week for the next 3 months. Each follow-up message will offer behavioural messages to support quitting; information about health risks of tobacco use and benefits of quitting and advice on withdrawal symptoms and coping strategies.

### Training of tobacco cessation counsellors delivering the intervention

Counsellors at UPHCs are allied healthcare workers and will be trained by the members of the research team to deliver counselling intervention. They will not be involved

in data collection or data analysis. Their training will be done using a standardised training manual (hardcopy of study manual will be provided to these counsellors) to maintain the fidelity of the intervention.

### Outcomes

The main outcomes expected from this exploratory trial include:

1. The proportion of all the tobacco users approached who consent to randomisation.
2. The proportion of those randomised who comply with the intervention.
3. The proportion of those randomised on whom follow-up data are collected at 3 months.

We will also assess the feasibility of assessing outcomes of primary interest in the main trial as listed below:

▶ 'Self-reported tobacco abstinence' of 7 days confirmed by a salivary cotinine test.
▶ Self-reported motivation and intention to quit.
▶ Cost-effectiveness as cost per disability-adjusted life year (DALY) averted.
▶ Cost-effectiveness as cost per quality-adjusted life year (QALY) gained.

The outcomes will be assessed at baseline on recruitment to the study and the end of 3 months. At baseline, sociodemographic details such as age, gender, educational level, marital status, religion, community, and economic status will be recorded. Variables related to tobacco consumption will be recorded, including the number of SLT products per day (frequency) consumed, age of initiation, number of previous cessation attempts and longest period of cessation. The data on quit attempts, the status of tobacco use and challenges will be recorded during baseline and end point. The biochemical validation of self-reported tobacco abstinence for 7 days will be assessed using salivary cotinine at the end of 3 months. The rapid test kit works through visual interpretation of colour developed on the kit and is instant in processing.[27 28] Participants will be approached by the trial team members to provide a small sample of saliva, which will be collected through passive drooling. Collection, transportation and disposition of the biological samples and the kit will be managed at the study site by the site coordinator and research team. Cotinine is a preferred marker for tobacco use because cotinine stays in the body much longer than nicotine. The presence of cotinine in saliva or urine is a reliable indicator of tobacco use. Cotinine in vivo has a half-life of about 20 hours and lasts for about 3–4 days in the saliva and urine.[29 30]

### Sample Size

Based on the primary outcomes in this exploratory trial of recruitment to the study and attrition, a total of 250 participants (125 participants per group) are required to estimate an anticipated proportion of 50% recruitment of participants with a 95% CI of 44% to 57% and 20% attrition at follow-up with a 95% CI of 15% to 25%. The sample size was calculated based on estimating proportions with a specified level of precision at the 95% level

as measured by the width of the CI using the Sample Size Tables for Clinical Studies software.[31]

## Randomisation and allocation

Randomisation will be done at the level of individual eligible participants who have provided informed consent. Randomisation will be stratified by study practice site (UPHCs) using random permuted blocks of varying block sizes from 4 to 10 and a 1:1 allocation to intervention or control arm. A randomisation list will be prepared by an independent statistician (from PRIMENT Clinical Trials Unit, University College London) not involved with this study. This list will be mailed to an independent staff member at the study site in India who will be responsible for the allocation of participants to the respective intervention arm. The independent staff member will assign a unique identification number to each of the consented participants and will maintain a list of those allocated to the respective study arm. The updated list will then be sent to the randomisation coordinator who will be responsible for overseeing the recruitment and randomisation process. The randomisation coordinator will remove all the participant identifiers from the list and finally send the list to the statistician for the purpose of analysis.

The trial will ensure a clear separation between staff who collect outcome data and those who deliver the intervention. Staff collecting outcome data will be blinded to the group assignment. None of the intervention staff, primary care physicians and behavioural counsellors will collect outcome data. All investigators, staff and participants will be kept masked to outcome measurements and any preliminary results emerging from the trial.

## Data management

Baseline and end-point data will be collected from participants at UPHC using surveys in paper format. This will be entered electronically in the database developed in MS Access (Microsoft Office 2019). The quality of the data will be monitored by a data supervisor for completeness, validity and integrity. In case of incomplete data or inconsistent responses, the data supervisor will verify these with the study participants and update the entries in the database. The database will incorporate a range of inconsistency checks to limit data entry errors. Time-stamped audit trail will also be built in the database to independently record the date and time of operator entries and actions that create, modify, or delete electronic records.

## Data analysis
### Statistical analysis

Participant characteristics will be summarised using mean and SD or median and IQR for continuous variables, and number and percentages for the categorical variables. The proportion of participants recruited and lost to follow-up at the end point will be estimated with 95% CIs. As part of the secondary analyses, logistic regression models will be used to estimate the intervention effects with 95% CIs for the prespecified outcomes, tobacco abstinence and self-reported motivation and intention to quit after adjusting for the stratification factor (UPHC) and baseline values of the outcome where it is available. These analyses will be done on an intention to treat (participants as randomised with available outcome data) basis, and multiple imputation will be used to handle missing outcome values if considered appropriate. The extent of missing data for each variable and the percentage of participants adhering to the intervention will be reported. Attrition levels by randomised group and the characteristics of participants who are lost to follow-up will also be reported. All analyses will be done using STATA software V.17 (StataCorp. 2019. *Stata Statistical Software: Release* V.17. College Station, Texas: StataCorp LLC) or updated versions. A detailed statistical analysis plan will be drawn up nearer to the analysis stage, prior to the database lock.

## Economic evaluation

The aim will be to evaluate the feasibility of calculating the cost-effectiveness of routine care along with intervention compared with routine care alone from an Indian healthcare cost perspective over 3 months. This will include the level of data completeness of self-completed questionnaires of tobacco-related healthcare resource use, estimation of the intervention and routine care costs. The feasibility of calculating cost per DALY averted, particularly in relation to potential cases of cancer avoided, as part of the main trial will also be evaluated in addition to data requirements for a decision model to evaluate the lifetime cost-effectiveness of the intervention calculated as the cost per QALY gained.

## Trial status

The two major COVID-19 pandemic waves have affected India (wave 1 in March 2020 and wave 2 in April 2021). This has delayed the recruitment of the trial since outpatient departments were not operational or operating with limited staff. Our current plan is to start recruitment by mid-July 2021. As per the original proposal, the follow-up was for a period of 6 months, our current plan for follow-up is now for 3 months.

## DISCUSSION

To the best of our knowledge, this is the first study evaluating a complex low-cost intervention of two components, that is, face-to-face counselling along with mobile phone messaging counselling delivered to SLT users in low resource Indian settings. The first component of the intervention is based on 5A's approach, which is known to be effective.[32] However, such a brief intervention may have a limited effect if the person is not ready to quit on account of other factors associated with difficulty in quitting tobacco.[33] However, follow-up quitting advice to tobacco users can increase quit attempts and quit rates,[32] and this will be achieved through the mobile phone messaging. Mobile-based interventions are widely used

in the developed world, and research on smoking cessation suggests that they can increase tobacco quitting by 25%–50%.[20] A study conducted in Sweden assessed the heterogeneous treatment effects of text messaging intervention among smokers and found that the effect was less pronounced among the participants with stronger nicotine dependence.[34] Tailoring of such interventions for specific individuals needs to be considered, as each individual will need messaging according to their beliefs and their readiness to quit. The mobile messages in our study will be developed with this context in mind. To date, there have been no evaluations conducted to assess the effectiveness of such complex low-cost intervention in low resource settings and primary care. The development and delivery of comprehensive, tailored primary care behavioural and mobile phone health technology interventions will enhance the understanding of low-cost long-term prevention intervention for tobacco cessation. Recent evidence has emerged on the association of tobacco use with novel coronavirus disease.[35 36] The use of SLT often involves hand-to-mouth contact that can promote the spread of COVID-19. Another behaviour that can spread infections associated with the use of SLT is the spitting of or projection of excess saliva produced during the chewing process.[37] While the struggle with coronavirus infection control continues, we have an opportunity to expand cessation intervention specifically designed for SLT use and promote good infection control practice.

## Strengths and limitations

This study has many strengths. The results of this study will inform the feasibility of conducting effectiveness trials in the future. Incorporating messaging technology in routine clinical practice will be time-efficient as the messages can be delivered anywhere, at fixed times, and directly to the participant with negligible direct contact while also ensuring privacy. The impact of this research on the training of health professionals will lead to an improved understanding of the delivery of low-cost tobacco cessation interventions in primary care. There are, however, some limitations to the study. As the study is being conducted in urban health centres in Berhampur city, Odisha, the findings would not be generalisable to other healthcare settings as well as to the whole population of the Odisha state. The primary care physicians may get unmasked to the allocation of the participants during the course of the trial as the participants visiting the UPHCs may reveal the information to the physicians. There is a chance of biased results if physicians may deliver repeated face-to-face counselling services. The inclusion criteria include SLT users having a mobile phone, this may exclude a section of the SLT users who do not have a personal mobile phone like people of low socioeconomic status or elderly who do not use a mobile phone. The follow-up is only for 3 months. Longer follow-up assessments might provide different findings.

## Research impact

In this study, SLT cessation intervention will be developed using an innovative method including a formative research phase, where the SLT cessation intervention will be developed; and the exploratory randomised controlled trial phase, where the evaluation of this intervention will be conducted. The evidence from this exploratory study will inform the acceptability and feasibility of SLT cessation interventions in users of SLT and will inform the conduct of a larger multicentric trial across several centres and countries. Findings from this study will provide insights into designing similar studies and appropriate interventions in tobacco cessation and noncommunicable diseases. Since research in this area is in its infancy in LMIC, this work will provide an impetus for researchers working on tobacco cessation to generate new evidence and allow them to adapt the tools developed and piloted in this study in real-time practice.

## Ethics and dissemination

Ethics approval has been obtained from the Institutional Ethical Committee at Public Health Foundation of India (PHFI) (ref: TRC-IEC-391/19; dated May 29, 2019). At the national level, ethical clearance has also been obtained from the Health Ministry's Screening Committee (HMSC), led by the Indian Council of Medical Research (ICMR) (ref: 2019–3581; dated December 11, 2019). Also, local level approval was obtained from the Odisha State Ethics Board (ref: 191/PMU/187/17; dated November 14, 2019). In the UK, ethical clearance was obtained by the UCL Research Ethics Committee (ref: 5686/001, dated October 1, 2019). The study has been registered at Clinical Trials Registry India (reference number CTRI/2019/05/019484). All the participants (participating in both the phases in this study) will be provided with a participant information sheet (PIS), providing details of the study. Following this, voluntary written consent for participation will also be taken from them (online supplemental appendix). All the data that will be collected for the study will be stripped of any personal identifiers and the data will be stored in PHFI's data repository. The data will only be accessible to the principal investigator and trial team analysing the data. To ensure confidentiality, data shared to project team members will be blinded for any identifying participant information. The findings of the study will be published in peer-reviewed journals.

**Contributors** All authors contributed to the manuscript. RP (PI of the study—India) and IN (PI of the study—UK) designed the study, led on writing, review and finalising the paper. RO provided statistical guidance. RH provided inputs on designing the economic evaluation section of the study. AM provided inputs on designing the formative research section of the study. RRP reviewed and assisted in writing the early drafts of the paper. All authors provided critical feedback and helped in shaping the manuscript.

**Funding** This research work is funded by the Joint Global Health Trials Initiative MRC/DfID/Wellcome Trust (ref: MR/P021166/1). This research work is sponsored by University College London, Gower Street, London, WC1E 6BT, UK.

**Competing interests** None declared.

**Patient consent for publication** Not applicable.

**Provenance and peer review** Not commissioned; externally peer reviewed.

**ORCID iDs**
Rachael Hunter http://orcid.org/0000-0002-7447-8934
Irwin Nazareth http://orcid.org/0000-0003-2146-9628

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
