## [Reviewer comments · BMJ Open]

ARTICLE DETAILS

TITLE (PROVISIONAL)	Exploratory randomized trial of face to face and mobile phone counselling against usual care for tobacco cessation in Indian primary care: a randomized controlled trial protocol for project CERTAIN.
AUTHORS	Panda, Rajmohan; Omar, Rumana; Hunter, Rachael; Prabhu, Rajath R; Mishra, Arti; Nazareth, Irwin

VERSION 1 – REVIEW

REVIEWER	Erin Vogel Stanford Prevention Research Center, Department of Medicine, Stanford University, Medicine
REVIEW RETURNED	21-Feb-2021

GENERAL COMMENTS	This paper describes the protocol for an exploratory randomized controlled trial of a mobile phone counselling tobacco cessation interview conducted in primary care clinics in Odisha, India. Below are some suggestions to improve clarity. • Throughout the paper, it is stated that the follow-up period will be dependent on the COVID-19 situation. The reason for this should be explained in the body of the paper, not just the “trial status” section. Until the “trial status” section, it is unclear that the COVID-19 situation impacts follow-up time by impacting the trial’s start date.• Page 2: The article summary states, “Evidence on the effectiveness of smokeless tobacco cessation intervention in primary care scares.” Should this state that “evidence is scarce”?• Page 3: “The majority of people use a variety of tobacco products – combustible, non-combustible, or both.” Please specify that this refers to the majority of people in India who use tobacco, not the majority of people in India.• Page 3: At the beginning of paragraph 4, it would be helpful to know what % of India’s population uses a mobile phone.• Page 4: Under “Aims and Objectives,” it is not clear that routine care does include brief tobacco counselling (i.e., ask and advise).• Page 4: The total duration of the study is 18 months. Please specify the duration of each phase.• Page 5: In Phase I, will participants also vary in readiness to quit tobacco (in addition to varying in demographics)?• Page 5: In Phase IIb, I’d like to know more about how participants will rate the newly developed messages using the validation tool. Are they rating the messages based on how much they like each message, or are there other criteria?• Page 5: The manuscript states that IDIs will be conducted with participants who dropped out of the study. Do the authors mean to say that IDIs will be conducted with participations who stopped engaging with the text messages? If they asked to be removed from the study, they should not be contacted.
---

	 • Page 6 (intervention components): Will the text messages be tailored to participants' readiness to quit? • Page 6: The manuscript states that if any participants have mental health pathology identified during counseling, treatment will be discontinued. Are there specific exclusion criteria linked to participants' ability to safely participate in the trial? Mental health pathology is quite common. Will participants be excluded even if the physician believes they can safely and effectively participate in the trial?
--	---

REVIEWER	Marcus Bendtsen Linköping University
REVIEW RETURNED	05-Mar-2021

GENERAL COMMENTS	1. Abstract  • Usual care is not defined, please mention what this is. • Is the recruited population all tobacco users, or only those who use smokeless tobacco? • In what way does follow-up depend on Covid-19? • The details of formative research can be removed (see later comments as well). • Please mention how findings will be disseminated. • Please provide a link to the trial registry, I couldn't find it when searching. 2. Strengths and limitations  • "scarc" should be "is scarce" • "such trials" should be "effectiveness trials" • The final limitation is not clear to me, please elaborate • In what way will it be time efficient given that additional work is added to current practices to the clinical work? 3. Introduction  • It would be helpful to quote mobile phone users as per cent of the population rather than absolute numbers. • Unclear interval 9% to 54%, please elaborate on included studies which are most relevant for current study 4. Aims and objectives  • I would remove objective 1, as it is not really an objective of the exploratory RCT itself, and then shorten the other objectives, so that it says: "Specific objectives of the exploratory trial are:" 1. Assess the proportion of all primary care tobacco..., etc. 5. Methodology This is a protocol for a randomized experiment which has a particular purpose (to avoid duplication and to ensure non-tampering of hypotheses and outcomes). However, authors have also added information about what they call "Phase I" formative research. I strongly suggest removing all the formative research descriptions (and the division of Phase I and Phase II) and focus this protocol only on the randomized experiment. 6. PPI  • The formative development could be mentioned here briefly. If authors prefer, the parts pertaining to Study phase I can be moved to an appendix. • Participant inclusion criteria for the formative studies should be mentioned if kept in appendix. 7. Study setting  • Is the slash ("/") meant to be understood as "and"? • Ethics can be moved to a section "Ethics and dissemination" as per BMJ Open instructions and SPIRIT. Please put this after the discussion. 8. Study population
--

	 • How is "current SLT users" defined (daily, monthly, past 8 weeks?) • Who will approach participants and how will eligibility be assessed? • How many per cent of the SLT user population is anticipated will visit a clinic each year? 9. Randomization  • The description of blinding is great, thank you. However, the process of allocating participants was a bit unclear, could this please be revised? • Is there a risk of disclosure bias at follow-up or during the allocation, please elaborate on this. 10. The arm of the study and Intervention  • Please rename this section to Interventions. • Reference to 5As approach is missing. • The information on routine care here is great, it would be good to add a short mention in the abstract. • The final sentence before Outcome measures (line 31-32) is redundant and can be removed. • How is personalisation of messages divided among participants (it says in Phase I that messages will be personalised) 11. Economic evaluation It would be good if authors could clarify their intentions and thinking around the economic evaluation. The aim of the research project is to increase overall health in the community by adding practices to the clinic, rather than comparing two equivocal interventions to decide which is superior. However, the economic evaluation seems to focus on the cost-effectiveness of the additional practices (which naturally will increase costs) in contrast to DALYs. Considering that few of all SLT users in the community will visit clinics, this focus may be too narrow. Would authors therefore consider focusing on the overall well-being of the community the clinic is serving by evaluating QALYs resulting from the additional practices? 12. Outcome measurements I would prefer if outcome measures can be made more prominent so that it is clear what the primary and secondary outcomes are. Data collected at baseline can be deferred to an appendix, as this section concerns the outcomes rather than baseline measurements.  • It is not clear what decides if follow-up is 3 or 6 months (it says dependent on Covid-19, but not how). • There seems to be a duplication of the heading Outcomes, so what I was looking for before now comes later. I would merge these two so that they both are before Data Management. • There seems to be two primary outcomes in one, both self-reported and saliva tests. Please clarify if these should be considered two separate outcomes. • It is unclear if the outcomes numbered 1,2,3 are part of the trial outcomes or which outcomes have precedence. 13. Sample size It is unclear to me why the sample size calculation has been based on the anticipated recruitment rate as an outcome when the primary outcome is tobacco cessation. This needs more elaboration, and perhaps becomes clearer once all outcomes have been listed together and priorities made clear. Also unclear why the attrition rate would matter if it is the recruitment rate which is the outcome, as recruitment rate cannot be subject to attrition. 14. Statistical analysis This section needs to be revised.  • How will 95% confidence intervals be estimated for recruitment rate and attrition? • Saying "applicable regression models" gives a license to use a
--	---

	vast array of different models.  • The type of regression used must be specified for each outcome, and variables used for adjustment needs to be specified. • Will any interaction models be analysed? • It needs to be specified if analyses will be intention-to-treat or per-protocol. • Inference methods needs to be explained (maximum likelihood or Bayesian?) • If any hypotheses will be tested, then thresholds need to be declared. • Will any imputation methods be used and how? • How will attrition be analysed, ensuring that attrition is not systematically different between arms, etc. 15. Discussion  • I am missing a discussion about study limitations and generalisability of results. • Authors might consider incorporating studies of heterogenous effects of the intervention as the population is very heterogenous and participants may react differently, see for instance: https://journals.plos.org/plosone/article/comments?id=10.1371/journal.pone.0229637 16. Other  • A SPIRIT figure is missing which would help to understand participant burden. • Are participants asked to come back to the unit for follow-ups, and will there be reminders? 17. Data sharing statement  • I think there is a lot of data generated for this study, so the statement isn't accurate.
--	---

VERSION 1 – AUTHOR RESPONSE

Reviewer Number	Original comments of the reviewer	Reply by the author (s)	Changes done on the page number and line number
1	Throughout the paper, it is stated that the follow-up period will be dependent on the COVID-19 situation. The reason for this should be explained in the body of the paper, not just the “trial status” section. Until the “trial status” section, it is unclear that the COVID-19 situation impacts follow-up time by impacting the trial's start date.	We have explained the uncertainty of the follow-up period in the ‘trial status’ section. This information is now under a new heading titled “trial status”, located at the end of the methodology section.	Pg.no. 7, Line no. 34-39
	Page 2: The article summary states, “Evidence on the effectiveness of smokeless tobacco cessation intervention in primary care scares.” Should	Yes, we did intend to say that “evidence is scarce”. This correction has been made.	Pg. no. 2, Line No. 24

	this state that “evidence is scarce”?		
	Page 3: “The majority of people use a variety of tobacco products – combustible, non-combustible, or both.” Please specify that this refers to the majority of people in India who use tobacco, not the majority of people in India.	We have rewritten the line in the context of tobacco users in India with a relevant reference - ‘The majority of the tobacco users in India use a variety of tobacco products – combustible, non-combustible, or both.’	Pg. no. 3, Line no. 5-6
	Page 3: At the beginning of paragraph 4, it would be helpful to know what % of India’s population uses a mobile phone.	The percentage of mobile phone users has been added to the Introduction section with a relevant reference.	Pg. no. 3, Line no. 28
	Page 4: Under “Aims and Objectives,” it is not clear that routine care does include brief tobacco counselling (i.e., ask and advise).	Routine care does include brief tobacco counselling (i.e., ask and advise). We have described the routine care under the Interventions section.	Pg. no. 5, Line no. 10-14
	Page 4: The total duration of the study is 18 months. Please specify the duration of each phase.	We have specified the duration for each phase under the study design section.	Pg. no. 4, Line no. 35-36
	Page 5: In Phase I, will participants also vary in readiness to quit tobacco (in addition to varying in demographics)?	Yes, the participants will also vary in readiness to quit tobacco.	-
	Page 5: In Phase IIb, I’d like to know more about how participants will rate the newly developed messages using the validation tool. Are they rating the messages based on how much they like each message, or are there other criteria?	Participants will rate the messages on a 10-point scale on the clarity and appeal of each message. In-depth interviews will be conducted with these participants to get their opinions on the developed messages to improve the format and content of the messages.	-
	Page 5: The manuscript states that IDIs will be conducted with participants who dropped out of the study. Do the authors mean to say that IDIs will be conducted with participations who stopped engaging with the text messages? If they asked to be removed from the study, they should not be contacted.	The participants who do not continue with the intervention will still be contacted to assess study outcomes and they will be asked why they did not partake of the intervention. We will not contact those participants who indicate that they do not want to be contacted when they withdraw from the study.	-

	Page 6 (intervention components): Will the text messages be tailored to participants' readiness to quit?	Yes, the text messages will be tailored to participants' readiness to quit. During the development stage, the messages will be developed based on participants' readiness to quit.	-
	Page 6: The manuscript states that if any participants have mental health pathology identified during counselling, treatment will be discontinued. Are there specific exclusion criteria linked to participants' ability to safely participate in the trial? Mental health pathology is quite common. Will participants be excluded even if the physician believes they can safely and effectively participate in the trial?	In the exclusion criteria, we have mentioned that "the participants who do not have the mental capacity to consent" will be excluded. By this, we mean that the participants who have dementia or learning disabilities and cannot provide informed consent will not be included in the study. We agree that mental health pathology is quite common among tobacco users and providing counselling to quit tobacco can have a direct positive effect on the mental health of the participants. Hence, line 54-59 on page no. 7 has been removed.	-
2	Abstract a. Usual care is not defined, please mention what this is. b. Is the recruited population all tobacco users, or only those who use smokeless tobacco? c. In what way does follow-up depend on Covid-19? d. The details of formative research can be removed (see later comments as well). e. Please mention how findings will be disseminated. f. Please provide a link to the trial registry, I couldn't find it when searching.	a. As suggested, a brief description of usual care has been included in the abstract section. b. Only smokeless tobacco users will be recruited to the exploratory RCT. We have clarified this in the abstract. c. This depends on the functioning of the normal OPDs in the urban health centres on account of the COVID 19 pandemic. We have expanded on this under a new heading titled "trial status" inserted at the end of the methods section of the paper. d. We have removed the formative research detail from the abstract section. e. We have mentioned the dissemination plan in the abstract section. f. The link to the trial registry has been added under the trial registration number heading in the abstract section.	a. Pg. no. 2, Line no. 12 b. Pg. no. 2, Line no. 9 c. – d. Pg. no. 2, Line no. 14 e. Pg. no. 2, Line no. 17-18 f. Pg. no. 2, Line no. 20-21
	Strengths and limitations a. "scares" should be "is scarce" b. "such trials" should be "effectiveness trials" c. The final limitation is not clear to me, please elaborate d. In what way will it be time efficient given that additional work is added to current practices to the clinical work?	a. "Scares" has been changed to "is scarce" in the strengths and limitation section. b. "Such trials" has been changed to "effectiveness trials" in the strengths and limitation section. c. The final limitation is that this study is based in Urban Public Health Centres (UPHCs). These are primary health care centres that cater to only a segment of the urban	a. Pg. no. 2, Line no. 24 b. Pg. no. 2, Line no.

	population. Hence, our findings won't be generalizable to other secondary or tertiary care settings (include district or sub-district hospitals) or private hospitals. d. No additional work is added. We are just strengthening the usual care. Smokeless tobacco users will be counselled with mobile health messages, and this will save time for the doctor and the counsellor.	25 c. – d. –
Introduction a. It would be helpful to quote mobile phone users as per cent of the population rather than absolute numbers. b. Unclear interval 9% to 54%, please elaborate on included studies which are most relevant for current study	a. The percentage of mobile phone users has been added to the Introduction section with a relevant reference. b. The interval has been changed from 9% to 51.5%. The Nethan et. al. (2018) study, which is a systematic review, included studies from 1966 to 2017. They assessed the efficacy of the intervention from the reported risk ratios (RRs) [confidence intervals] and quit rates. They found the quit rates to between 9-51.5%, at six months.	a. Pg. no. 3, Line no. 28 b. Pg. no. 3, Line no. 41
Aims and objectives a. I would remove objective 1, as it is not really an objective of the exploratory RCT itself, and then shorten the other objectives, so that it says: "Specific objectives of the exploratory trial are:" 1. Assess the proportion of all primary care tobacco..., etc.	The first objective has been removed and the other objectives have been shortened.	Pg.no. 4, Line no. 13-21
Methodology This is a protocol for a randomized experiment which has a particular purpose (to avoid duplication and to ensure non-tampering of hypotheses and outcomes). However, authors have also added information about what they call "Phase I" formative research. I strongly suggest removing all the formative research descriptions (and the division of Phase I and Phase II) and focus this protocol only on the randomized experiment.	We have removed all of "Phase I" formative research from the body of the manuscript and we now focus on the protocol of the randomised trial in this manuscript.	Pg.no. 4
PPI a. The formative development could be mentioned here briefly. If authors prefer, the parts pertaining to Study phase I can be moved to an	a. As suggested, we have provided an overview of the formative development in the PPI section and have moved the parts pertaining about study phase I to Appendix. (Appendix A)	a. Pg.no. 4, Line no. 23-31

	appendix. b. Participant inclusion criteria for the formative studies should be mentioned if kept in appendix.	b. The participant inclusion criteria for formative study have also been mentioned in the appendix A.	b. Supplementary file - Appendix A
	Study setting a. Is the slash ("/") meant to be understood as "and"? b. Ethics can be moved to a section "Ethics and dissemination" as per BMJ Open instructions and SPIRIT. Please put this after the discussion.	a. No, the slash "/" in this context means "or". We have removed the term "primary care clinics" to avoid confusion. b. As suggested, the Ethics section has been moved to the "Ethics and dissemination" section after the discussion.	a. Pg. no. 4, Line no. 39 b. Pg. no. 9, Line no. 8-22
	Study population a. How is "current SLT users" defined (daily, monthly, past 8 weeks?) b. Who will approach participants and how will eligibility be assessed? c. How many per cent of the SLT user population is anticipated will visit a clinic each year?	a. The "current Smokeless tobacco users" for this study are defined as the Smokeless tobacco users who have used them in the last three months. b. A separate data collection team (only responsible for screening, informed consent collection, baseline, and end-point data collection) will approach the participants. The eligibility of the participants will be assessed based on the inclusion/exclusion criteria for the participants and a screening log will also be maintained during assessment of eligibility. c. We anticipate 30-40% of the Smokeless tobacco users visit a clinic each year.	a. Pg. no. 4, Line no. 43 b. - c. -
	Randomization a. The description of blinding is great, thank you. However, the process of allocating participants was a bit unclear, could this please be revised? b. Is there a risk of disclosure bias at follow-up or during the allocation, please elaborate on this.	a. As suggested, we have elaborated on the process of allocation of the participants in the randomisation section. b. There won't be any disclosure bias during allocation as randomisation and allocation of the participants will be done by an independent staff member who is not a part of the trial. During the follow-up as well, there won't be any disclosure bias as the participants will be recruited from 9 different UPHCs and from different communities as well.	a. Pg.no. 6, Line no. 32-39 b. -
	The arm of the study and Intervention a. Please rename this section to Interventions. b. Reference to 5As approach is missing. c. The information on routine	a. The section has been renamed "Interventions". b. The reference for 5A has added. c. Some information on routine care has been added to the abstract. d. The sentence before Outcome measures has been removed	a. Pg. no. 5, Line no. 5 b. Pg. no. 11,

	care here is great, it would be good to add a short mention in the abstract. d. The final sentence before Outcome measures (line 31-32) is redundant and can be removed. e. How is personalisation of messages divided among participants (it says in Phase I that messages will be personalised)	e. In the phase I-formative study, after collecting the data from different stakeholders, we will develop the messages based on the Transtheoretical Model of behaviour change that assumes that individuals move through six stages of change: pre-contemplation, contemplation, preparation, action, maintenance, and termination. The messages will be tailored based on the person's stages of change. This will allow for personalised messages.	Line no. 11-13 c. Pg. no. 2, Line no. 12 d. Pg.no. 5, Line no. 33 e. –
	Economic evaluation It would be good if authors could clarify their intentions and thinking around the economic evaluation. The aim of the research project is to increase overall health in the community by adding practices to the clinic, rather than comparing two equivocal interventions to decide which is superior. However, the economic evaluation seems to focus on the cost-effectiveness of the additional practices (which naturally will increase costs) in contrast to DALYs. Considering that few of all SLT users in the community will visit clinics, this focus may be too narrow. Would authors therefore consider focusing on the overall well-being of the community the clinic is serving by evaluating QALYs resulting from the additional practices?	The aim of the economic evaluation is to evaluate the feasibility of conducting a full economic evaluation comparing the CERTAIN intervention to routine care. As a result, it will predominately focus on reporting descriptive statistics and not calculating a summary statistic. To provide additional information about what we would consider as the aim of the economic evaluation as part of a full trial though, given that Smokeless tobacco is associated with a range of negative health problems, the long-term aim of the intervention is to reduce the risk of the negative health impact of Smokeless tobacco, including the risk of head and neck cancers. DALYs were considered more suitable given (i) their role in decision making in middle income countries such as India, where a decision threshold for QALYs might not be as suitable; and (ii) that they provide a method for quantifying avoiding multiple poor health outcomes such as multiple different cancers and cardiovascular disease. Since the aim of this study is an exploration of the options we have included an exploration of QALYs as suggested by the reviewer as well as clarifying the use of DALYs.	Pg. no. 7, line no. 26-33
	Outcome measurements I would prefer if outcome measures can be made more prominent so that it is clear what the primary and secondary outcomes are. Data collected at baseline can be deferred to an appendix, as this section concerns the outcomes rather than baseline measurements. a. It is not clear what decides if follow-up is 3 or 6 months (it	This has been revised. As per the SPIRIT checklist, under the outcome section (#12) we have mentioned the specific measurement variables, analysis metric, method of aggregation, and time point for each outcome. Hence, the time points of data collection are listed under the outcome section and not presented as an appendix. a. The follow-up depends on the normal functioning of the OPDs in the UPHCs. Due to the first and second COVID-19 wave in	

	says dependent on Covid-19, but not how). b. There seems to be a duplication of the heading Outcomes, so what I was looking for before now comes later. I would merge these two so that they both are before Data Management. c. There seems to be two primary outcomes in one, both self-reported and saliva tests. Please clarify if these should be considered two separate outcomes. d. It is unclear if the outcomes numbered 1,2,3 are part of the trial outcomes or which outcomes have precedence.	India, the OPDs in the UPHCs were shut down and were operating with limited staff. This had an impact on the start of recruitment of the study participants. b. The duplication of the heading Outcomes has been removed and the sections have been merged. c. In outcomes, we will also assess the feasibility of the outcomes. We will assess participant's self-reported tobacco abstinence of seven days confirmed by the salivary cotinine test. The salivary cotinine test is a validation test that will be done for confirmation of self-reported tobacco abstinence. d. This study is an exploratory trial to assess the feasibility of the study i.e., to assess if we can develop a main multicentre trial and assess the effectiveness of the complex intervention developed from this study on a larger population. Hence, the outcomes numbered 1,2,3 are the main outcome for this exploratory trial and the primary & secondary outcomes are to check for the feasibility of the intervention under this trial. Also, we have removed the heading of the primary and secondary objectives to make the outcomes clearer.	a. – b. Pg. no. 5, Line no. 33 c. – d. Pg.no. 6, line no. 1-4
	Sample size It is unclear to me why the sample size calculation has been based on the anticipated recruitment rate as an outcome when the primary outcome is tobacco cessation. This needs more elaboration, and perhaps becomes clearer once all outcomes have been listed together and priorities made clear. Also unclear why the attrition rate would matter if it is the recruitment rate which is the outcome, as recruitment rate cannot be subject to attrition.	The primary outcomes for the exploratory trials are the recruitment and attrition rates of participants and not tobacco cessation, which will be the primary outcome for the subsequent definitive trial. The attrition rate will measure the proportion of people who will be lost to follow-up at three-months post recruitment, or the proportion of patients retained in the trial at three-months follow-up. The statistical approach to calculating sample size estimation for pilot and exploratory studies used in the paper are referenced (Sample Size Tables for Clinical Studies, 3rd Edition Wiley [Internet]. Wiley.com. [cited 2020 Oct 7]. Available from: https://www.wiley.com/enus/Sample+Size+Tables+for+Clinical+Studies%2C+3rd+Edition-p-9781444357967)	Pg.no. 6, line no. 21-26
	Statistical analysis This section needs to be	a) The 95% CIs for proportions will be estimated using the formula that is used to calculate CIs for single proportions assuming normality for large samples. The	Pg. no. 7, line no. 11-

	revised. a. How will 95% confidence intervals be estimated for recruitment rate and attrition? b. Saying "applicable regression models" gives a license to use a vast array of different models. c. The type of regression used must be specified for each outcome, and variables used for adjustment needs to be specified. d. Will any interaction models be analysed? e. It needs to be specified if analyses will be intention-to-treat or per-protocol. f. Inference methods needs to be explained (maximum likelihood or Bayesian?) g. If any hypotheses will be tested, then thresholds need to be declared. h. Will any imputation methods be used and how? i. How will attrition be analysed, ensuring that attrition is not systematically different between arms, etc.	formula which is derived from, introductory statistics books i.e.: for level $1-\alpha$, the CI is: Estimated Proportion $\pm Z_{\alpha/2} \times (\text{Std. Error. of the estimated proportion})$, where $Z_{\alpha/2}$ is the $1 - \alpha/2$ quantile of the standard normal and $Z_{\alpha/2}$ s equal to 1.96 for 95% CIs. The Standard Error of a proportion is estimated by the formula: $\text{Sqrt}[P \times (1 - P)/n]$ where P is the estimated proportion. Reference Stata software manual: StataCorp. 2019. Stata 16 Base Reference Manual. College Station, TX: Stata Press. b) & c) The choice of the regression model will depend on the outcome for example, linear, logistic or Poisson to analyse continuous, binary and count outcomes. This has been clarified in the manuscript. We have also indicated that analysis will be adjusted for the baseline values of the outcome. d) We do not perceive the need to include interaction terms in the model. However, in discussion with trials team, should this issue arise, it will be specified in the detailed statistical analysis plan which will be drawn up later, prior to analysis and before the database lock. e) We have specified that the analysis for the secondary outcomes will be intention to treat. f) The inference methods will be specified in the detailed statistical analysis plan; it is most likely to be based on frequentist rather than Bayesian methods. g) We are not testing any statistical hypotheses in the exploratory trial. The main outcomes of interest are recruitment and follow up of study participants as would be expected from an exploratory study. h) We have added a sentence regarding imputation. This will be described in more detail in the statistical analysis plan. i) This is an exploratory trial where overall attrition levels, attrition levels by randomized group and characteristics of patients who were lost to follow-up will be reported.	25 b & c. Pg. no. 7, line no. 17-18 d. - e. Pg no. 7, line no. 18-20 f. - g. -
--	---	---	--

			h. Pg. no. 7, line no. 19
			i. Pg. no. 7, Line no. 21-23
	Discussion		
	a. I am missing a discussion about study limitations and generalisability of results. b. Authors might consider incorporating studies of heterogenous effects of the intervention as the population is very heterogenous and participants may react differently, see for instance: https://journals.plos.org/plosone/article/comments?id=10.1371/journal.pone.0229637	a. The strengths and limitations of the study have been added to the discussion section. b. Thank you for the suggestion, we have incorporated information about heterogeneous effects of the intervention in the discussion section along with the relevant reference.	a. Pg. no. 8, Line no. 20-29 b. Pg. no. 8, Line no. 5-7
	Other		
	a. A SPIRIT figure is missing which would help to understand participant burden. b. Are participants asked to come back to the unit for follow-ups, and will there be reminders?	a. We have prepared a SPIRIT figure showcasing the phases of trial and data collection time points. b. Yes, the participants will be asked to come back to the unit for follow-ups. In both arms, after the participants' baseline assessment and following delivery of the intervention (i.e. - only routine care (in control arm) and routine care + face-to-face intervention (in the intervention arm), participants will be contacted to return to the clinic for the assessment of the endpoint and salivary cotinine test. There will be reminders and their travel costs will be covered for this visit	a. Table 1, Pg. 12 b. –
	Data sharing statement		
	a. I think there is a lot of data generated for this study, so the statement isn't accurate.	a. We have revised the data sharing statement for this study protocol.	Pg.no. 9, Line no. 23-25

VERSION 2 – REVIEW

REVIEWER	Marcus Bendtsen Linköping University
REVIEW RETURNED	30-Sep-2021

GENERAL COMMENTS	The protocol is improved. I still have two concerns which you may or may not want to address, as it would strengthen your study significantly. 1. The section on statistical analysis still lacks detail. You are compromising the quality of your study by not explicitly saying that outcome 1 will be analysed using regression model X adjusted for Z,W and will be reported using W... Saying that you will create a plan later is weakening the study. The point of publishing a protocol is to reduce duplicate efforts and to reduce tampering of methods post-hoc. I leave it in the hands of the editor to decide if the detail suffices. 2. There are many limitations when conducting these types of trial, some which are often unavoidable. However, it seems that you haven't identified a single one. You mention generalisability, but that isn't a limitation per se (all trials are generalisable to a point). It is concerning that you haven't raised a single limitation, as it conveys that you haven't spent time thinking about what may bias your results, and haven't put efforts towards reducing these biases. Please note that discussing limitations of your trial strengthens the study, not the other way around - the quality of your trial will be significantly improved if you can identify potential sources for bias and address them, or at least acknowledge that there is a risk and how your results may be affected.
---

VERSION 2 – AUTHOR RESPONDS

Reviewer Number	Original comments of the reviewer	Reply by the author (s)	Changes done on the page number and line number
2	The protocol is improved. I still have two concerns which you may or may not want to address, as it would strengthen your study significantly. The section on statistical analysis still lacks detail. You are compromising the quality of your study by not explicitly saying that outcome 1 will be analysed using regression model X adjusted for Z,W and will be reported using W... Saying that you will create a plan later is weakening the study. The point of publishing a protocol is to reduce duplicate efforts and to reduce tampering of methods post-hoc. I leave it in the	We have revised the “Statistical Analysis” section and have added specific analysis and regression methods.	Pg.no. 7, Line no. 11-25

	hands of the editor to decide if the detail suffices.		
	There are many limitations when conducting these types of trial, some which are often unavoidable. However, it seems that you haven't identified a single one. You mention generalisability, but that isn't a limitation per se (all trials are generalisable to a point). It is concerning that you haven't raised a single limitation, as it conveys that you haven't spent time thinking about what may bias your results, and haven't put efforts towards reducing these biases. Please note that discussing limitations of your trial strengthens the study, not the other way around - the quality of your trial will be significantly improved if you can identify potential sources for bias and address them, or at least acknowledge that there is a risk and how your results may be affected.	We have revised the "strengths and limitations" section of the manuscript	Pg. no. 2, Line no. 23-37; Pg. no.8, Line no. 20-35